# Relationship between Sleep Duration Trajectories and Self-Rated Depressive Symptoms in South Koreans with Physical Disabilities

**DOI:** 10.3390/healthcare9030361

**Published:** 2021-03-23

**Authors:** Su Jeong Yi, Yoo Mi Jeong, Jae-Hyun Kim

**Affiliations:** 1College of Nursing, Dankook University, 119 Dandae-ro, Dongnam-gu, Cheonan-si 31116, Chungchungnam-do, Korea; 12181056@dankook.ac.kr; 2Department of Health Administration, College of Health Science, Dankook University, 119 Dandae-ro, Dongnam-gu, Cheonan-si 31116, Chungchungnam-do, Korea

**Keywords:** sleep trajectory, depressive symptoms, physical disability

## Abstract

Physically disabled persons can have sleep problems, which affects their mental health more than those in non-disabled people. However, there are few studies on the relationship between sleep duration and mental health targeting physically disabled people in South Korea, and existing studies on the disabled have mostly used data collected from convenience rather than nationally representative samples, limiting the generalization of the results. This study used data from the second wave of the Panel Survey of Employment for the Disabled (PSED, 2016–2018, 1st–3rd year). Participants included 1851 physically disabled individuals. The Chi-square test and generalized estimating equation (GEE) were used and the Akaike information criterion (AIC) value and the AIC log Bayes factor approximation were used to select sleep trajectories. This is the first study to elucidate multiple sleep trajectories in physically disabled people in Korea, and the relationship between sleep duration trajectories and self-rated depressive symptoms. People with physical disabilities who sleep more than 9 h have the highest risk of depression and need more intensive management as a priority intervention.

## 1. Introduction

About 15% of the world’s population experiences some form of disability, of which 2–4% have significant functional difficulties [1]. The number of registered disabled persons in South Korea as of the end of December 2019 was 2.62 million accounting for about 5.1% of the overall population [2]. The registered population of disabled persons was 1.22 million in 2019, accounting for 47% of the overall population, of which 97.7% were attributed to acquired causes [2]. Traffic accidents are a major cause of physical disabilities, resulting in visible damage [3]. Unexpected accidents can leave an individual psychologically scarred [4], leading to a negative psychological state [5]. According to the Korea Employment Agency for Persons with Disabilities [2], 59.1% of people with physical disabilities appeared stressful consistently, and 16.1% reported feeling hopeless for more than two weeks in the past year, affecting daily routine. Since people with disabilities are more likely to experience associated health and psychological problems, compared with the non-disabled, continuous health management is imperative [6]. 

At the population level, people with disabilities registered poor mental health [7] and were also at a high risk [8] compared to the non-disabled. People with disabilities may experience negative emotions, such as depression, loneliness, anger, suicidal tendencies, etc. [9,10]. The prevalence of depression among persons with disabilities is 17.03% compared with 7.83% for non-disabled persons [6,9]. Individuals with disabilities are more vulnerable to depression [11], and three to four times more prone to depression [12] than non-disabled people. Accordingly, people with disabilities not only experience high levels of depression due to their disability [13], but depression itself negatively affects their psychology, job, lifestyle, and happiness [14]. People with depression have low self-esteem [15] and suicidal tendencies [16]. In people with disabilities, depression worsens physical health and disability, negatively affecting their own disability acceptance and rehabilitation will [17]. In addition, depression accompanies physical, cognitive, and social problems, such as worsening of their quality of life and suicide, and increases social loss and burden [18]. Depression among people with disabilities affects their physical, mental, and economic aspects [19]. Therefore, it is necessary to explore factors affecting depression among the disabled and draft an appropriate strategy to improve mental health.

It has been recognized that adequate sleep is very important for maintaining mental health [20]. Sufficient sleep refers to a state where you are no longer sleepy when you wake up, and you sleep to the extent that you do not fall asleep on your own while awake [21]. Sleep duration has a curvilinear association with a wide range of health outcomes [22] and sleep quality is closely associated with mental health disorders. Short or long sleep has been studied in mental health disorders, and the relationship between sleep disturbance and mental disorders has been reinforced to be bidirectional [23]. In addition, there were mixed results about the relationship between sleep duration and depression regarding whether only sleep deprivation or both sleep deprivation and oversleeping impacts on the risk of depression [20,24,25]. Sleep deprivation was found to have a significant effect in the model without control variables but disappeared in the model corrected for employment status and other related factors [26]. Moreover, a study by Hwang and Park [27] showed that suicidal tendencies increased 1.33 times in adults with a shorter sleep duration than those with proper sleep. Therefore, examination of sleep duration and depression with a longitudinal data analysis is required in order to identify the relationship between them clearly in people with physical disability. 

Accordingly, inadequate sleep is related to mental problems, and sleep is considered an important factor affecting mental health, especially for the disabled. However, studies related to the sleep duration of disabled people showed that they are more likely to have sleep problems than non-disabled people [22,28], although there are few studies on the relationship between sleep duration and mental health. Existing studies on the disabled mostly used data collected from convenience rather than nationally representative samples, limiting the generalization of the results. A longitudinal understanding is essential for understanding the sleep time and mental health of the disabled in a multidimensional manner, but since most studies are cross-sectional, it is difficult to identify causality. Therefore, this study aimed to analyze longitudinal data to increase causality and examine how depression among persons with physical disabilities is affected differently, according to sleep duration.

## 2. Methods

### 2.1. Study Sample and Design

This study used data from the second wave of the Panel Survey of Employment for the Disabled (PSED, 2016–2018, 1st–3rd year) conducted by the Korea Employment Agency for the Disabled. About 4577 registered disabled persons stipulated in Article 2 of the Welfare Act for Persons with Disabilities aged 15–64 residing nationwide were included as of May 15, 2016. Variations in age and disability type were considered, and those who wished for economic activity were over-allocated.

PSED provides panel data from repeatedly measured households comprising persons with disabilities that include the demographic and socioeconomic characteristics of individuals, factors related to disabilities, and household variables (e.g., income and expenditure, including medical costs). The survey was conducted every year between May and July and involved face-to-face interviews with persons with disabilities, using a computer-assisted personal interviewing (CAPI) method for collecting accurate information. The participants entered their responses in the computer-installed CAPI and the investigators could check logically incorrect responses. If a direct response was impossible due to an intellectual disability or mental disorder, the head of the household or nearest guardian was allowed to respond. To estimate the association between sleep duration and depression among physically disabled individuals, 1851 participants were included with no missing information.

### 2.2. Independent Variables

Sleep duration referred to self-reported data responding to the question “How many hours do you usually sleep?” Responses were assigned to one of five subcategories: ≤4 h, 5 h, 6 h, 7 h, 8 h, 9 h, and ≥10 h. The authors used the International Classification of Sleep Disorders 2nd edition definitions of sleep ≤ 4 h as “extremely short sleep,” 5 h as “short sleep,” 9h as “long sleep,” and ≥10 h as “extremely long sleep”.

### 2.3. Dependent Variables

#### Self-Rated Depressive Symptoms

Self-reported data regarding depressive symptoms were extracted from responses to the question: “Have you ever had feelings of being sad, blue, or depressed for two weeks or more during the past year?” The presence of self-rated depressive symptoms was categorized into either “yes” or “no”.

### 2.4. Control Variables

The participants were classified into male and female, and categorized into the age groups: 15–29, 30–39, 40–49, 50–59, and ≥60 years. Educational level was categorized into four groups: elementary school or lower, middle school, high school, and college or higher. Marital status was divided into three groups: single, married, and separated. Residential region was categorized into metropolitan (Seoul), urban (Daejeon, Daegu, Busan, Incheon, Kwangju, or Ulsan), and rural. Smoking and alcohol consumption were categorized into three groups―never, former, and current. Disability level was either severe (level 1 to 3) or light (level 4 to 6), and disability type included physical and others, based on the number of samples of the disabled. Chronic disease other than disability included comorbidities associated with cancer, rheumatoid arthritis, gastritis, chronic hepatitis, diabetes, thyroid disease, hypertension, cardiovascular disease, cerebrovascular disease, tuberculosis, chronic bronchitis, asthma, cataract, chronic otitis media, and chronic renal failure; and self-rated health was categorized into good or bad. Employment status included worker in use, temporary, daily, self-employed, and unpaid family volunteer.

### 2.5. Analytical Approach and Statistics

The Chi-square test and generalized estimating equation (GEE) model were used to investigate the association between sleep duration and mental health among physically disabled individuals. Further, this study employed group-based trajectory modeling to identify distinctive trajectories of sleep duration for the entire sample. Trajectory modeling provides a method by which we can develop a probable representation of unobserved group classification and group differences, based on observed information and user-specified constraints. Once the sleep duration trajectories were derived from the trajectory modeling for identifying homogeneous subpopulations within the larger heterogeneous population based on the Akaike information criterion (AIC) value to select distinguishable trajectories and for describing longitudinal change within each unobserved sub-population, the trajectory classes were then coded into a series of dummy variables to examine the relationship between the patterns of sleep duration over time and mental health using the GEE models. For all analyses, the criterion for statistical significance was *p* < 0.05, two-tailed. They were conducted using the SAS statistical software package, version 9.4 (SAS Institute Inc., Cary, NC, USA).

## 3. Results

### 3.1. Prevalence of Self-Rated Depressive Symptoms

Table 1 displays the descriptive statistics of all variables at baseline (2016). Of the 1851 subjects, the prevalence of depression was 15.5% (287 participants) (Table 1). Additionally, 2.5% (46 participants) had an extremely short sleep duration, of which 28.3% (13 participants) had self-rated depressive symptoms, and 7.8% (144 participants) had a short sleep duration, of which 28.5% (41 participants) had self-rated depressive symptoms. Meanwhile, 2.2% of the sample (41 participants) had an extremely long sleep duration, of which 36.6% (15 participants) had self-rated depressive symptoms, and 2.9% (53 participants) had a long sleep duration, of which 18.9% (10 participants) exhibited these symptoms.

### 3.2. Association between Sleep Duration and Depression among Physically Disabled People

Table 2 was adjusted for socioeconomic and health status, and risk behavior variables. After adjusting for all these confounders, the odds ratio (OR) of self-rated depressive symptoms of those with an extremely short sleep duration was 1.751 times higher (*p* = 0.027) than those with 7 h of sleep. After adjusting for all confounders, the OR of self-rated depressive symptoms of those with an extremely long sleep duration was 4.179 times higher (*p* < 0.001) than those with a sleep duration of 7 h (Table 2). 

### 3.3. Changes in Sleep Duration over Time and Self-Rated Depressive Symptoms among the Physical Disabled

Figure 1 displays the sleep duration trajectory groups among the physically disabled population. A four-group linear solution for trajectories of sleep duration fit the data best in the trajectory analysis. This model had limited information criteria values (AIC, BIC, sample-size adjusted BIC) relative to other group solutions. Trajectory group 1 (4.0% of the sample) was characterized by the highest levels of sleep duration with a slight constant over time. Trajectory group 2 (50.7% of the sample) had appropriately constant levels of sleep duration, while trajectory groups 3 and 4 had low levels of sleep duration, with 8.6% and 36.7% belonging to this class, respectively (Figure 1). 

The sleep duration trajectory groups were used as dummy variables in the GEE models to examine the relationship between sleep duration patterns over time and depressive symptoms (shown in Table 3). Trajectory group 2 (those with appropriately constant sleep duration) was the reference category. In the adjusted model with the sleep duration trajectory group, those belonging to class 1 had the highest risk (OR: 2.073, *p*-value < 0.0001) and class 3 had a significant risk (OR: 1.503, *p*-value: 0.006) of depressive symptoms among all the sleep duration trajectory groups (Table 3).

## 4. Discussion

To overcome the limitations of existing cross-sectional studies, this study examined sleep duration trajectories longitudinally using the second wave of PSED, which included a substantial number of people with disabilities. 

It was found that 15.5% of people with physical disabilities had self-rated depressive symptoms, lower than the 18.4% derived from the Korea Employment Agency for Persons with Disabilities [2]. This difference can be attributed to the time of the survey and the subjects involved. Specifically, participants in this survey were mostly physically impaired, except for the sensory impaired, with less cognitive impairment and the highest social participation rate [14], which were considered to have a relatively low depression rate. In addition, the study showed that 5.1% of participants reported oversleeping for 9 h or more, and sleep deprivation for less than 6 h was reported by 10.3% of adults. Although direct comparison is difficult, less than 6.9% of participants reported more than 9 h of sleep and 15.1% reported sleeping for less than 6 h [29]. In a national study of people with physical disabilities, 69.2% reported sleeping for less than 7 h and 2.6% more than 8 h [6]. Considering the sleep duration difference, the number of sleep deprivation groups has decreased and oversleep groups has increased. However, in this study, only people with physical disabilities were included. The number of people with physical disabilities in the preceding study [6] was 39, and due to the small number, it was difficult to make specific conclusions, although it was possible to examine the patterns of sleep duration. Sleep time varies from country to country as well as according to population characteristics [29], and the risk of sleep problems varies based on the type of disability [22,28]. It is imperative to conduct a longitudinal study of sleep for people with physical disabilities in South Korea, since such people have difficulty maintaining a certain arousal cycle and controlling the circadian rhythm due to the problem of arousal control necessary for sleep [30].

In the regression analysis of sleep time and depression, self-rated depressive symptoms were 1.75 times higher among those with less than 4 h of sleep than those with 7 h of sleep. This is in line with the results showing that the depression score was lower in individuals with more than 8 h to less than 10 h than those with less than 8 h of sleep [31], although direct comparison is impossible because there is no study about the relationship between sleep duration and depression among people with disabilities. This is similar to the results that showed the highest depression in women with under 5 h of sleep [32]. Furthermore, it was different from the results in which the depression OR was significantly lower in 7–8.9 h and over 9 h compared to 7 or less hours of sleep time [20]. Self-rated depressive symptoms were 4.19 times higher when sleeping time was 10 h or longer compared to 7 h. This showed that unlike in the previous study, not only sleep deprivation but also oversleeping has a positive effect on depression. This is similar to an earlier finding that depression was highest when sleep duration was less than 4 h, and when the sleep during was 7–8.9 h, there were fewer self-rated depressive symptoms than that of 7 h or less or 9 h or more [25]. In South Korea, studies related to sleep duration and depression mainly includes adolescents [33,34], adults [20,27,32], and the elderly [35,36]. Studies on the disabled are limited, and most emphasize the importance of adequate sleep considering only the relationship between sleep deprivation and depression [20,27,32,33,34]. However, in a meta-analysis of prospective studies [37], a short and long sleep duration was significantly associated with an increased risk of depression in adults. Therefore, it is necessary to get at least 7–8 h of sleep. In particular, in the case of physical disorders, restricted movement is related to oversleeping. Physical activity is beneficial in reducing the risk of depression by promoting neurotransmitters, such as dopamine and serotonin, increasing the secretion of endorphins, and distracting from stress stimulation [37]. However, in the case of a person with a physical disability, restricted movement and long sleep duration further curtail physical activity. Therefore, it is necessary to study sleep patterns according to mobility and reinforce educational content on appropriate movement. In the general population, most previous studies showed that there was no consistent result, such as low depression OR [20] and high depression [25], in the case of excessive sleep of over 9 h. It is believed that further studies are needed to highlight this relationship and it is necessary to accumulate evidence through repeated studies in the future.

Four patterns were identified while examining the sleep time trajectory of individuals with physical disabilities over time. In group 1, the oversleeping group, it was found that the sleep duration gradually decreased to 8 h, consistent with a study by Gilmour et al. [38] on general populations, who found that the short sleep time group showed a slight downward liner trend. Conversely, there was no significant change in groups 3 and 4, where sleep time increased over time, which might be due to the non-linear association between sleep and depression in the long sleep group as they showed a quadratic trend in the previous study [38]. These results may not have clinical significance, and it is difficult to conclude that the trajectory group classification accurately reflected the change in sleep time of each individual assigned to the trajectory. Therefore, a longer follow-up period may be required to determine meaningful changes in sleep time.

Regarding the influence of sleep patterns on self-rated depressive symptoms, in group 1, the OR of self-rated depressive symptoms is 2.07 times higher than that of the reference group, which registered 7 h of sleep, although the sleeping time gradually increased to 8 h. It is positive that the sleeping time decreased from 9.6 h in 2016 to 8 h in 2018, but the higher probability of falling into depression than the group that maintained 7 h of sleep means that it is important to sleep for 7–8 h for more than two years. In the preceding study [39], most of the participants maintained an adequate sleep time for 20 years, ensuring a healthy lifestyle [40]. Sufficient or 7–8 h of sleep is imperative for a healthy and productive lifestyle, and people who sleep for less or more hours are at risk of adverse health effects [41]. Sufficient sleep is essential for restoration and recovery [42], and for maintaining mental health [20]. Accordingly, people who maintain an adequate sleep pattern over a long period are less prone to depression. 

The OR of depression was 1.50 times higher in group 3, the minimum sleep group, compared to the reference group, supporting the findings that both short and long sleep increased depression [25]. The mechanism by which too short or long sleep times increase the risk of depression due to genetic [25] or biological factors [43] is not yet clear, but physical exercise was found to alleviate sleep problems and depression in persons with disabilities [44]. In addition, people with low levels of physical activity, compared with those with high levels, had a lower OR of depression [45]. Since people with disabilities are more likely to have sleep problems than non-disabled people [22,28], appropriate interventions are important to maintain proper sleep duration for more than a year. It is necessary to identify sleep time in the intervention and prevention of depression for persons with physical disabilities and to maintain appropriate sleep time by mediating sleep disorder factors. In particular, people with physical disabilities who sleep more than 9 h have the highest risk of depression and need more intensive management as a priority intervention. It is imperative for future studies to specifically verify which factors trigger differences in the occurrence of depression between groups. Moreover, according to previous studies, there was a difference in sleep and depression according to the type of circadian rhythm [46,47,48]. The physical circadian rhythm directly affects psychological and behavioral variables, such as mood, sleepiness, and mental concentration, and affects the sleep–wake cycle, hormone secretion, body temperature, and other important body functions [49]. Almost all physiological functions and behaviors of our body follow the circadian rhythm [50], and the human body can maintain optimal health status by harmonizing changes in and outside the body through the circadian rhythm [51]. However, in this study, it was difficult to understand the circadian rhythm because this data did not distinguish between daytime sleep and nighttime sleep; thus, the role of the circadian rhythm between sleep duration and depression needs to be examined. 

Our study also showed that factors influencing depression were marital status, current smoking, disability level, chronic disease, employment, and self-rated health. These were consistent with results that showed that depression differed based on smoking [52,53] and employment status [13]. Meanwhile, gender [54,55,56,57], age [55,56], and alcohol consumption [58,59] influenced depression, although these factors were not associated with depression in this study. This can be attributed to the diverse subjects and classification criteria for each study. In this study, the ratio of men and w omen was 3:1, and in the case of drinking, it is believed that the amount or duration of alcohol consumption was not considered, and thus the results varied. Therefore, it is necessary to examine variables that do not appear as factors explaining depression in this study through future studies.

The limitations of this study are as follows. First, there were limitations in extracting data on not only depression but also suicide, a risk factor, along with extracting trajectory influence factors by using secondary data. Therefore, in understanding the results of this study, these limitations should be considered, and if possible, more relevant factors should be included in subsequent studies. Second, depression was measured by a piecemeal question of “Do you feel depressed?” without using depression screening tools. In addition, in measuring sleep time, objective results are insufficient using individual reports on how many hours a day the person sleeps. In addition, it is necessary to measure the actual time through Actiwatch rather than just subjective data on sleep duration. Therefore, it is necessary to secure higher reliability by using objective tools or physiological variables when understanding depression and sleep in persons with physical disabilities. Third, self-examination is retrospective data, so there is a possibility that there might be recall bias. Fourth, the classification of trajectory groups is probabilistic, so individuals belonging to each trajectory do not accurately follow the pattern of sleep time [38]. Lastly, although sleep disorders may have an effect on sleep time and depression, the original data were not considered as there were no results for the clinical diagnosis of sleep disorders. Therefore, the clinical diagnosis of sleep disorders should be included in the future research to examine more clearly the relationship between sleep duration and depression. However, since the derivation of sleep duration trajectories from longitudinal data provides more information than cross-sectional as it can identify patterns over time, these findings are considered to provide adequate information for subsequent studies related to sleep in people with physical disabilities. 

## 5. Conclusions

Although the disabled have a high mental health risk, in-depth studies on the subject are limited. This study attempted to analyze the influence of sleep duration on self-rated depressive assessment among people with physical disabilities using representative PSED data. Consequently, four sleep duration trajectories were identified, and among them, people with physical disabilities who sleep more than 9 h have the highest risk of depression and need more intensive management as a priority intervention. Although it is too early to generalize the analysis results due to the absence of similar existing studies, the significance of this study is to share basic research results and to provide direction for future sleep and depression studies of disabled people. It is imperative that the relationship between sleep time and depression is examined among people with physical disabilities, including more diverse variables, considering both sleep quality as well as sleep duration. 

## Figures and Tables

**Figure 1 healthcare-09-00361-f001:**
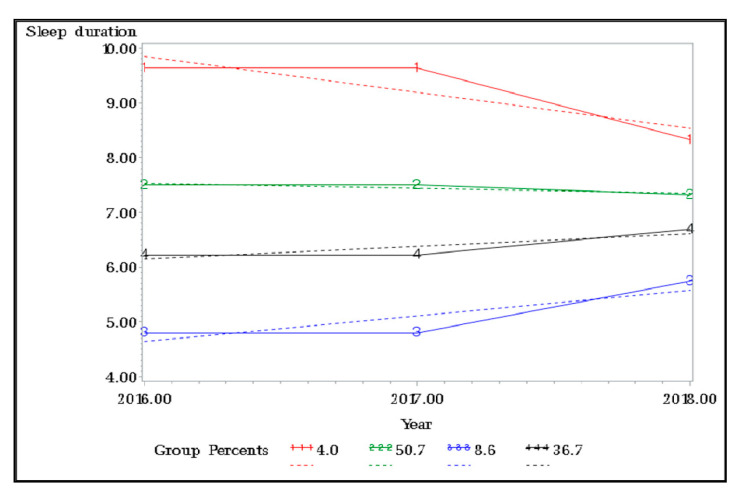
The sleep duration trajectory groups among the people with physical disabilities.

**Table 1 healthcare-09-00361-t001:** The descriptive statistics of all variables at baseline (2016).

All Variable at Baseline	Total		Self-Rated Depressive Symptom				*p*-Value
			Yes		No		
	*N*	%	*N*	%	*N*	%	
Sleep duration (h)							<0.001
≤4	46	2.5	13	28.3	33	71.7	
5	144	7.8	41	28.5	103	71.5	
6	465	25.1	73	15.7	392	84.3	
7	656	35.4	73	11.1	583	88.9	
8	446	24.1	62	13.9	384	86.1	
9	53	2.9	10	18.9	43	81.1	
≥10	41	2.2	15	36.6	26	63.4	
Gender							0.211
Male	1277	69.0	189	14.8	1088	85.2	
Female	574	31.0	98	17.1	476	82.9	
Age							0.000
15–29	140	7.6	18	12.9	122	87.1	
30–39	445	24.0	50	11.2	395	88.8	
40–49	688	37.2	100	14.5	588	85.5	
50–59	367	19.8	83	22.6	284	77.4	
≥60	211	11.4	36	17.1	175	82.9	
Residential region							0.568
Metropolitan	292	15.8	49	16.8	243	83.2	
Urban	535	28.9	76	14.2	459	85.8	
Rural	1024	55.3	162	15.8	862	84.2	
Marital status							<0.001
Married	1087	58.7	115	10.6	972	89.4	
Single	473	25.6	85	18.0	388	82.0	
Divorce, separated	291	15.7	87	29.9	204	70.1	
Smoking status							0.030
Current smoker	537	29.0	102	19.0	435	81.0	
Former smoker	415	22.4	58	14.0	357	86.0	
Nothing	899	48.6	127	14.1	772	85.9	
Alcohol consumption							0.029
Drinker	974	52.6	135	13.9	839	86.1	
Former drinker	353	19.1	70	19.8	283	80.2	
Nothing	524	28.3	82	15.7	442	84.4	
Disability grade							0.001
1–3	388	21.0	81	20.9	307	79.1	
4–6	1463	79.0	206	14.1	1257	85.9	
Chronic disease							<0.001
Yes	479	25.9	116	24.2	363	75.8	
No	1372	74.1	171	12.5	1201	87.5	
Self-rated health							<0.001
Bad	858	46.4	212	24.7	646	75.3	
Good	993	53.7	75	7.6	918	92.5	
Employment							<0.001
Worker in use	818	44.2	75	9.2	743	90.8	
Temporary worker	266	14.4	30	11.3	236	88.7	
Daily worker	34	1.8	3	8.8	31	91.2	
Self-employed	70	3.8	16	22.9	54	77.1	
Unpaid family volunteer	663	35.8	163	24.6	500	75.4	
Total	1851	100.0	287	15.5	1564	84.5	

**Table 2 healthcare-09-00361-t002:** The OR of self-rated depressive symptom among the people with physical disabilities.

Sleep Duration	Self-Rated Depressive Symptom			
	*OR*	95% CI		*p*-Value
Sleep duration (h)				
≤4	1.751	1.065		2.878
5	1.845	1.346		2.530
6	1.106	0.869		1.407
7	1.000			
8	1.265	1.008		1.586
9	2.703	1.736		4.207
≥10	4.179	2.616		6.675
Gender				
Male	0.868	0.672		1.121
Female	1.000			
Age				
15–29	0.738	0.457	1.194	0.216
30–39	1.153	0.829	1.603	0.399
40–49	1.173	0.890	1.546	0.256
50–59	1.112	0.847	1.460	0.444
≥60	1.000			
Residential region				
Metropolitan	0.864	0.674	1.107	0.247
Urban	0.764	0.624	0.935	0.009
Rural	1.000			
Marital status				
Married	0.461	0.370	0.575	<0.001
Single	0.763	0.586	0.995	0.046
Divorce, separated	1.000			
Smoking status				
Current smoker	1.497	1.140	1.966	0.004
Former smoker	1.201	0.897	1.608	0.218
Nothing	1.000			
Alcohol consumption				
Drinker	0.904	0.711	1.151	0.412
Former drinker	0.932	0.718	1.210	0.595
Nothing	1.000			
Disability grade				
1–3	1.485	1.222	1.805	<0.001
4–6	1.000			
Chronic disease				
Yes	1.445	1.191	1.753	0.000
No	1.000			
Employment status				
Worker in use	0.435	0.348	0.545	<0.001
Temporary worker	0.497	0.373	0.662	<0.001
Daily worker	0.184	0.057	0.596	0.005
Self-employed	0.857	0.528	1.393	0.534
Unpaid family volunteer	1.000			
Self-rated health				
Bad	2.329	1.893	2.865	<0.001
Good	1.000			
Perceived stress				
Yes	3.149	2.566	3.865	<0.001
No	1.000			
Year				
2016	1.346	1.091	1.659	0.006
2017	1.116	0.900	1.385	0.317
2018	1.000			

**Table 3 healthcare-09-00361-t003:** The relationship between sleep duration patterns overtime and self-rated depressive symptom.

Sleep Patterns Overtime	Self-Rated Depressive Symptom			
	*OR*	95% CI		*p*-Value
Sleep patterns overtime				
1	2.073	1.460	2.944	<0.001
2	1.000			
3	1.503	1.124	2.011	0.006
4	1.170	0.966	1.417	0.108
Gender				
Male	0.913	0.709	1.178	0.485
Female	1.000			
Age				
15–29	0.754	0.468	1.215	0.246
30–39	1.174	0.846	1.630	0.337
40–49	1.161	0.883	1.527	0.286
50–59	1.108	0.846	1.452	0.455
≥60	1.000			
Residential region				
Metropolitan	0.898	0.702	1.149	0.394
Urban	0.777	0.636	0.949	0.014
Rural	1.000			
Marital status				
Married	0.456	0.366	0.567	<0.001
Single	0.757	0.582	0.984	0.038
Divorce, separated	1.000			
Smoking status				
Current smoker	1.481	1.131	1.941	0.004
Former smoker	1.181	0.884	1.576	0.261
Nothing	1.000			
Alcohol consumption				
Drinker	0.876	0.690	1.113	0.279
Former drinker	0.926	0.715	1.199	0.559
Nothing	1.000			
Disability grade				
1–3	1.482	1.221	1.798	<0.001
4–6	1.000			
Chronic disease				
Yes	1.430	1.181	1.732	0.000
No	1.000			
Employment status				
Worker in use	0.400	0.320	0.500	<0.001
Temporary worker	0.469	0.353	0.624	<0.001
Daily worker	0.170	0.053	0.550	0.003
Self-employed	0.818	0.505	1.324	0.413
Unpaid family volunteer	1.000			
Self-rated health				
Bad	2.388	1.944	2.933	<0.001
Good	1.000			
Perceived stress				
Yes	3.067	2.504	3.756	<0.001
No	1.000			
Year				
2016	1.371	1.114	1.687	0.003
2017	1.130	0.913	1.400	0.261
2018	1.000			

## Data Availability

Data available in a publicly accessible repository that does not issue DOIs Publicly available datasets were analyzed in this study. This data can be found here: https://www.kead.or.kr/common/comm_board_v.jsp?no=435&gotopage=1&search=4&keyword=&data_gb=007&branch_gb=B01&station_gb=000&main=4&sub1=4&sub2=0&sub3=0&option (accessed on 5 January 2021).

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
