# Peer review of "Relationship between Sleep Duration Trajectories and Self-Rated Depressive Symptoms in South Koreans with Physical Disabilities"

_healthcare, 2021, doi:10.3390/healthcare9030361_

Round 1
Reviewer 1 Report
Jeong Yi et al. analyze a large longitudinally profiled cohort of disabled individuals and correlate depressive symptoms to sleep debt. The study is of importance considering the lack of representation amongst disabled individuals in routine sleep related studies. The authors aptly list their limitations which is a welcome change, especially the subjectivity of the study and use of actigraph devices to better track sleep schedules. Its difficult to uncouple the mutual effects of sleep and depression, does depression lead to sleep debt or lack of sleep leads to depression. I only have minor concerns
- The method used to group sleep trajectories is not entirely clear. More details could help the reader understand the conclusions. Did any of the individuals who improved their sleep debt decrease their depressive symptoms.
- Did any of these individuals had clinically diagnosed depression? if so did you exclude them from analyses because of treatment?
- Can the authors speculate about the role of circadian rhythm in promoting depression?
Author Response
- The method used to group sleep trajectories is not entirely clear. More details could help the reader understand the conclusions. Did any of the individuals who improved their sleep debt decrease their depressive symptoms.
- Thank you for your comments. We added more details about group sleep trajectories in method section.
2. Did any of these individuals had clinically diagnosed depression? if so did you exclude them from analyses because of treatment?
- Thank you for your comments. Unfortunately, we used the existed data collected from disabled association, and there was no such clear data about clinically diagnosed depression. Therefore, we included all disabled people who reported any depressive symptoms.
3. Can the authors speculate about the role of circadian rhythm in promoting depression?
- Thank you for your comments. We added more explanation about circadian rhythm in discussion section and suggested its application in future study.

Reviewer 2 Report
This longitudinal study investigates on the relationship between sleep duration and mental health in 1,851 physical disabled people in South Korea.
As main finding, the study reports that people with physical disabilities who sleep more than 9 hours have the highest risk of depression.
This study has strenghts and some weaknesses.
Strenghts: sample size and longitudinal measures
Weaknesses: sleep duration and depresion were self-reported and measured by a single question.
The authors have to:
- Clearly state in their discussion that the measure of sleep duration is based on a single question andf not by a standard measure (like the Pittsburgh Sleep Quality Index)
- Across the whole manuscript "depressive symptom" should be replaced by "self-rated depressive assessment"
- Another major limitation is the lack of any information on the presence of sleep disorders. This is a basic limit of the investigation, since all results could point to relationships that have been due to sleep disorders and not to sleep duration, as estimated a unique question. This point should be deepenly discussed
- Please, improve the quality of the figure
Author Response
1. Clearly state in their discussion that the measure of sleep duration is based on a single question and if not by a standard measure (like the Pittsburgh Sleep Quality Index)
--> Thank you for your comments. We definitely agree with you about using the standard measure, but this data was not included such measures. Therefore, we added this explanation in limitation part.
2. Across the whole manuscript "depressive symptom" should be replaced by "self-rated depressive assessment"
--> Thank you for your comments. We revised it as you recommended.
3. Another major limitation is the lack of any information on the presence of sleep disorders. This is a basic limit of the investigation, since all results could point to relationships that have been due to sleep disorders and not to sleep duration, as estimated a unique question. This point should be deepenly discussed
--> Thank you for your comments. We added this explanation in discussion section.
4. Please, improve the quality of the figure
--> Thank you for your comments. We improved the quality of the figure.

Round 2
Reviewer 2 Report
Tha authors responded to the points I raised, with a notable exception. In fact, they did not change in the title and in the abstract "depression" with "self-rated depressive symptoms.
As they made across the whole manuscript, this term should be changed accordingly also in the title and abstract.
Afrter this change, the manuscipt will be accptable for publication
Author Response
The authors responded to the points I raised, with a notable exception. In fact, they did not change in the title and in the abstract "depression" with "self-rated depressive symptoms. As they made across the whole manuscript, this term should be changed accordingly also in the title and abstract.
After this change, the manuscript will be acceptable for publication
--> Thank you for your comments. We revised it accordingly in the title and abstract.
